# Modulation of Epstein-Barr-Virus (EBV)-Associated Cancers by Co-Infections

**DOI:** 10.3390/cancers15245739

**Published:** 2023-12-07

**Authors:** Christian Münz

**Affiliations:** Viral Immunobiology, Institute of Experimental Immunology, University of Zürich, Winterthurerstrasse 190, 8057 Zürich, Switzerland; christian.muenz@uzh.ch

**Keywords:** Epstein Barr virus (EBV), Kaposi-sarcoma-associated herpesvirus (KSHV), malaria, *Plasmodium falciparum*, human immunodeficiency virus (HIV), human cytomegalovirus (HCMV), human papilloma virus (HPV), *Helicobacter pylori*, Burkitt’s lymphoma, primary effusion lymphoma (PEL)

## Abstract

**Simple Summary:**

Epstein Barr virus (EBV) is a constant companion of humans with more than 95% of the adult population being persistently infected. Fortunately, this companionship is mostly asymptomatic, but in around 300,000 new patients every year, EBV-associated cancers emerge. These medical cases are mostly caused by environmental or genetic conditions that either increase EBV-associated tumorigenesis or weaken its immune control. In this review, I will discuss co-infections as important environmental risk factors for EBV-associated malignancies and how both EBV infection but also the respective co-infections could be targeted to reduce the associated disease load.

**Abstract:**

The oncogenic and persistent Epstein Barr virus (EBV) is carried by more than 95% of the human adult population. While asymptomatic in most of these, EBV can cause a wide variety of malignancies of lymphoid or epithelial cell origin. Some of these are also associated with co-infections that either increase EBV-induced tumorigenesis or weaken its immune control. The respective pathogens include Kaposi-sarcoma-associated herpesvirus (KSHV), *Plasmodium falciparum* and human immunodeficiency virus (HIV). In this review, I will discuss the respective tumor entities and possible mechanisms by which co-infections increase the EBV-associated cancer burden. A better understanding of the underlying mechanisms could allow us to identify crucial features of EBV-associated malignancies and defects in their immune control. These could then be explored to develop therapies against the respective cancers by targeting EBV and/or the respective co-infections with pathogen-specific therapies or vaccinations.

## 1. Introduction

With more than 95% of the human adult population being persistently infected with the Epstein Barr virus (EBV), this human γ-herpesvirus is one of the most successful pathogens in humans [1,2,3]. Despite its potent growth-transforming abilities of human B cells in culture to lymphoblastoid cell lines (LCLs) [4], EBV causes only around 300,000 new malignancies per year [5,6]. Curiously, these virus-associated tumors are quite heterogenous in presentation and cellular origin, including nasopharyngeal carcinoma of epithelial cell origin, Burkitt’s and Hodgkin’s lymphoma of B cell origin and NK/T cell lymphomas that emerge from lymphocytes that are not found to be EBV-infected under non-pathological conditions [7]. Apart from the different cellular origin of EBV-associated malignancies, differences in the viral gene expression program causes some of this heterogeneity. While Burkitt’s lymphomas usually express only one vial protein, the nuclear antigen 1 of EBV (EBNA1) in so-called EBV latency I, Hodgkin’s lymphoma expresses EBNA1 and the two latent membrane proteins (LMP1 and 2) in latency IIa [2]. All EBNAs (1, 2, 3A, 3B, 3C and LP) and LMPs (1 and 2) are expressed in latency III, a program that is found in diffuse large B cell lymphomas (DLBCL) often associated with immune suppression. During non-pathological EBV persistence, all these latency programs can also be found in distinct B cell differentiation stages, namely latency III in naive B cells, latency II in germinal center B cells and latency I in homeostatically proliferating memory B cells [8,9]. In addition, EBV-infected resting memory B cells express only non-translated viral RNAs but no viral proteins (latency 0), and shortly after the EBV infection of B cells, the EBNAs without the LMPs are expressed (latency IIb) [10,11]. For latency 0 and I, lytic EBV replication for infectious virion production can be initiated by B cell receptor cross-linking and is associated with in vivo plasma cell differentiation [12]. Accordingly, co-infection with the Kaposi-sarcoma-associated virus (KSHV) that occurs in the very same cancer cells of primary effusion lymphoma (PEL) and drives plasma cell differentiation [13] results in increased lytic EBV gene expression [14]. Thus, all healthy EBV carriers harbor B cells with the viral expression programs of EBV-associated malignancies, and immune control prevents the transition to lymphomas.

This becomes apparent during immune suppression by, for example, the co-infection with human immunodeficiency virus (HIV) or after transplantation [15,16]. A more fine-grained picture of the molecular requirements for EBV-specific immune control originates from primary immunodeficiencies that predispose one for EBV-associated pathologies [17,18]. These individual mostly loss-of-function gene mutations point towards cytotoxic lymphocytes but not antibodies as being required for EBV-specific immune control. They identify genes in the differentiation of natural killer (NK) and T cells, in T cell receptor signaling, in the co-stimulation of cytotoxic lymphocytes, in their cytotoxic machinery as well as in their capacity to expand and survive [19]. Some of this information has been mechanistically interrogated in mice with reconstituted human immune systems (humanized mice) [20]. The depletion of NK cells and CD8^+^ T cells compromises the immune control of EBV in these in vivo models [21,22]. Furthermore, the co-stimulatory molecules 2B4 and CD27 are required for EBV-specific immune control in humanized mice [23,24]. Accordingly, EBV-specific T cells have proven clinically efficacious against EBV-associated malignancies upon the adoptive transfer into patients [25]. In particular, EBNA1- and LMP2-specific T cells have shown clinical efficacy without other T cell specificities [26,27,28]. However, it still remains unclear to what extent CD4^+^ T cells are needed for EBV-specific immune control. MHC-class-II-deficient patients seem to sufficiently control EBV [29], but compromising IL-2 production, primarily by CD4^+^ T cells, with the immune suppressant tacrolimus (FK506) or the depletion of CD4^+^ T cells due to HIV infection increases EBV-associated pathologies [15,30,31,32]. Nevertheless, a catalogue of features for a healthy EBV-specific immune control has been assembled and can be used to compare vaccine-induced immune responses that are developed for prophylaxis or therapy against EBV-associated diseases [33,34,35]. The modulation of the above-described malignancy-associated infection programs and/or of the immune control of EBV by co-infections that are often associated with EBV positive tumors need, however, to be taken into account, and may also be targeted to reduce the incidence of EBV-associated pathologies. These will be discussed next.

## 2. Human Immunodeficiency Virus (HIV)

HIV infection causes acquired immunodeficiency due to the gradual destruction of its main host cell compartment, helper CD4^+^ T cells [36]. The virus-induced loss of CD4^+^ T cell help gradually erodes adaptive immune responses, including the immune control of EBV-associated lymphomagenesis (Table 1) [15]. Accordingly, EBV-associated lymphomas are some of the hallmark diseases for acquired immunodeficiency syndrome (AIDS). Prior to antiretroviral therapy (ART), these were dominated by DLBCLs, and similar EBV-latency-III-expressing tumors increased in frequency in HIV and EBV co-infected humanized mice [15,32]. The depletion of helper CD4^+^ T cells due to HIV infection is associated with increased inhibitory co-receptor expression in cytotoxic CD8^+^ T cells, including PD-1, LAG3, TIM-3 and TIGIT [32,37]. This phenotype is associated with reduced functionality of CD8^+^ T cells upon the loss of CD4^+^ T cell help due to HIV infection (Figure 1A) and referred to as T cell exhaustion. Accordingly, CD8^+^ T cell depletion does not increase EBV viral loads during HIV and EBV co-infection compared with EBV infection alone in humanized mice [32]. The restoration of CD4^+^ T cell help due to ART reduced the DLBCL incidence in HIV-infected individuals [15]. Similarly, primary central nervous system (CNS) lymphomas (PCNSL) that are nearly uniformly EBV-infected significantly decreased in incidence after the introduction of ART. These findings indicate that HIV co-infection compromises EBV-specific immune control for highly immunogenic (all latent EBV proteins expressed as in latency III) lymphomas to develop.

However, for EBV-latency-I- and II-associated lymphomas, such as Burkitt’s lymphoma, Hodgkin’s lymphoma and primary effusion lymphoma (PEL), the role of HIV co-infection is more complicated. They develop also at increased frequencies in HIV-co-infected individuals [15]. Interestingly, their incidence has, however, not decreased, and for Burkitt’s and EBV-associated Hodgkin’s lymphoma, they have even increased after the introduction of ART. Therefore, they now compose around half of the EBV-associated lymphomas in people living with HIV (PLWH) [15]. Burkitt’s and Hodgkin’s lymphomas are thought to emerge from germinal centers [7], a site where HIV may still replicate during ART treatment [38]. The inflammatory environment with HIV-derived antigen stimulation may promote the development of EBV-associated lymphoma cells, which on their own are already more prone to mutations due to the induction of activation-induced deaminase due to EBNA3C [39]. One of these mutations is c-myc translocation into the immunoglobulin loci that is thought to depend on AID, which may be intrinsically cell-induced by EBV and extrinsically by HIV-infection-associated B cell stimulation via viral antigens. However, HIV may even cell-intrinsically increase the mutational load in EBV-infected B cells that can be infected by CXCR4 tropic HIV strains after EBV-infection-induced CD4 up-regulation [32]. The cytidine deaminases of the APOBEC3 family are induced by HIV infection and known restriction factors of retroviruses [40]. At the same time, cytidine deaminases like AID and APOBEC3 can also induce mutations in cancer cells [41,42]. Therefore, HIV and EBV may both contribute to oncogenic mutations in germinal-center B cells, from which both Burkitt’s and Hodgkin’s lymphomas are thought to originate. Thus, HIV co-infection may promote EBV-associated lymphomas both by immune suppression and by increasing oncogenic mutations in germinal centers (Figure 1A). 

## 3. Malaria and *Plasmodium falciparum*

EBV was discovered in 1964 in Burkitt’s lymphoma cells from Uganda [43,44]. In Uganda and other Sub-Saharan African countries, endemic Burkitt’s lymphoma continues to constitute one of the most frequent childhood tumors [45]. A similarly high Burkitt’s lymphoma incidence in Papua New Guinea suggests a geographical overlap with *Plasmodium falciparum* (*P. falciparum*) co-infection. Accordingly, measures or mutations, such as those causing sickle cell anemia that restricts *P. falciparum* infection, decrease Burkitt’s lymphoma incidences [46,47,48]. Apart from a nearly uniform infection of EBV, endemic Burkitt’s lymphoma is characterized by translocations of the cellular oncogene c-myc into one of the immunoglobulin loci. In most cases, Burkitt’s lymphoma is associated with EBV latency I, expressing non-translated BHRF1 and BART miRNAs and EBERs with EBNA1 being the only expressed protein [49]. However, there is a subset of Burkitt’s lymphomas in which only EBNA2 expression is inactivated by mutations [50]. Thus, endemic Burkitt’s lymphoma, which nearly uniformly harbors EBV without EBNA2 expression, is characterized by c-myc translocation and occurs at high frequencies in regions with *P. falciparum* infection but is rare in geographical regions with other *Plasmodium* species. 

In areas of high-*P. falciparum* co-infection, children carry high EBV loads [51]. In part, this may be due to *P. falciparum* compromising EBV-specific immune control. Along these lines, *P. falciparum*-infection-associated Burkitt’s lymphoma has been shown to be associated with terminally differentiated CD56^−^CD16^+^ NK cells [52,53] (Figure 1B), while EBV expands early differentiated CD56^+^CD16^+/−^NKG2A^+^ cells that restrict lytic EBV replication [22,54]. Moreover, decreased EBV-specific T cell responses were reported in children that were frequently exposed to *P. falciparum* infection [55,56,57]. This may be due to a shift to Th2-polarizing T cell priming conditions that are needed for optimal immune control of the blood stage of *P. falciparum* [58]. Indeed, Burkitt’s lymphoma development occurs in children with efficient parasitic immune control [59]. In addition to the alteration of EBV-specific immune control, *P. falciparum* may also directly increase EBV replication. Along these lines, it was shown that hemin, as the oxidized form of heme, released from infected erythrocytes, drove plasma cell differentiation and viral lytic replication in latently EBV-infected B cell lines [60]. Hemin seems to bind to BACH2, relieving its block or plasma cell differentiation and associated induction of the viral lytic cycle. Furthermore, *P. falciparum*-infected erythrocytes bind via erythrocyte membrane protein 1 (PfEMP1) to B cells and cause their activation, leading to EBV-lytic-reactivation-associated plasma cell differentiation [61]. This increased lytic EBV replication may both promote the suppression of EBV-specific immune control via viral IL-10 (BCRF1) [62] and generate higher frequencies of Burkitt’s lymphoma precursors that, via EBNA3C-mediated AID expression, may allow for the c-myc translocation that is characteristic of Burkitt’s lymphoma (Table 1) [39]. Moreover, *P. falciparum*-infected erythrocyte extract induces AID in B cells and hemozoin that are contained in this extract, which, together with B cell receptor cross-linking, also induces AID [63] (Figure 1B). The colonization of the erythrocytic lineage and the superior adhesion function of *P. falciparum*-infected erythrocytes are also among the main differences between Burkitt’s-lymphoma-associated *P. falciparum* and *P. vivax* that cause malaria in South America and Indo-Asia [64,65]. *P. falciparum* causes extensive erythrocyte infection (around 10% in severe malaria), while *P. vivax* infects reticulocytes, causing only around 1% of erythrocytic cells in the peripheral blood to be positive for this parasite [64,66]. Thus, *Plasmodium falciparum* infection is associated with EBV-associated endemic Burkitt’s lymphoma development in Sub-Saharan Africa and Papua New Guinea, weakening EBV-specific immune control and inducing elevated viremia by lytic cycle reactivation as well as stimulating expression of the mutagenesis machinery that is required for the c-myc translocation that is characteristic of Burkitt’s lymphoma.

## 4. Kaposi-Sarcoma-Associated Herpesvirus (KSHV)

Kaposi-sarcoma-associated herpesvirus (KSHV) is the second human γ-herpesvirus and was discovered in 1994 in Kaposi sarcoma, an AIDS defining malignancy [67]. Soon after, it was also identified in B cell malignancies, such as primary effusion lymphoma (PEL) and multicentric Castleman’s disease (MCD), and more recently in the osteosarcomas of Asian patients [68,69]. It is thought that KSHV persists in Igλ-expressing B cells that are enriched in PEL and MCD [70], but the reservoir of viral infections in healthy carriers has not been well-characterized. Persistent KSHV infection seems to be facilitated by environmental factors because seroprevalence is high in equatorial regions, such as Sub-Saharan Africa and in HIV-infected individuals worldwide, especially men who have sex with men [71,72]. The co-transmission of KSHV with EBV and/or HIV, as has been experimentally shown for homologous macaque viruses [73], could in part be responsible for the difference in seroprevalence. Accordingly, EBV transmission occurs nearly uniformly very early in life in Sub-Saharan Africans, and also, KSHV seroprevalence is increased if the mother carries this virus [74,75]. This results in nearly all KSHV carriers being EBV-positive with EBV co-infection being the main environmental risk factor for KSHV seropositivity [76,77,78]. Indeed, such a co-infection of B cells with EBV and KSHV led also to KSHV persistence in the resulting LCLs and in B cells of humanized mice [14,79,80]. A subset of these EBV-infected B cells carries KSHV [14,79]. Co-infection with EBV leads only to KSHV persistence in B cells when both infections occur at roughly the same time [79]. Interestingly, EBV and KSHV co-infection can also be observed in the majority of PELs (Table 1) [81]. Removing EBV from co-infected PEL cell lines leads also to the loss of KSHV [82]. Indeed, the co-infection of EBV and KSHV seems to increase lymphomagenesis in humanized mice compared with that in single-infected animals [14,80]. Tumor cell lines that can be grown from double-infected humanized mice show hallmarks of PEL, in particular, a plasmablastic cell differentiation [13,14]. The increased lymphomagenesis of KSHV and EBV co-infection depends in part on lytic EBV replication, which is induced by KSHV co-infection [14] (Figure 1C). The co-infection of KSHV with BZLF1-deficient-EBV that cannot switch into lytic replication causes a similar lymphoma incidence in humanized mice as does a single-EBV infection [14,80]. This KSHV-induced lytic EBV replication is probably facilitated by EBV latency I, which is found in most PELs, and their plasma cell differentiation that is also associated with lytic EBV replication in healthy virus carriers [7,12]. Thus, the co-infection of EBV and KSHV allows for KSHV persistence in B cells and contributes to PEL development. 

In addition, and similar to HIV and *P. falciparum*, KSHV influences EBV-specific immune control. This includes the differentiation of NK cells to terminally differentiated CD56^−^CD16^+^ cells [80]. While early differentiated CD56^+^CD16^+/−^NKG2A^+^ NK cells expand during primary EBV infection and target lytic EBV replication [22,54], KSHV co-infection with EBV causes an accumulation of CD56^−^CD16^+^KIR^+^CXCR6^+^ NK cells [80] (Figure 1C). These terminally differentiated NK cells carry the cytotoxic machinery (perforin and granzymes) but are unable to efficiently degranulate in response to susceptible target cells. Moreover, their antibody-mediated depletion did not influence viral loads in EBV and KSHV-co-infected humanized mice [80]. The adaptive immune response to KSHV in healthy carriers of both γ-herpesviruses is usually weaker than that to EBV. Frequencies of KSHV-specific T cell responses are usually 5- to 10-fold lower than those against EBV in the peripheral blood of healthy virus carriers [83]. In addition to these low responses, no immunodominance of KSHV antigens can be detected for T cells with 20 to 40% of donors responding to LANA (ORF73) and K8.1 as the most frequently recognized T cell antigens [83]. These are also among the most frequently recognized targets for antibody responses in KSHV-positive individuals, but also ORF65, ORF38, ORF61, ORF59 and K5 elicit robust antibody responses [84]. In addition to the significant difference in magnitude in EBV- versus KSHV-specific adaptive immune responses in co-infected individuals, KSHV-specific T cell immunity may also depend on different co-stimulatory and effector molecules that are identified by primary immunodeficiencies [17]. Apart from T cell receptor signaling, OX40, IFN-γR1 or STAT4 mutations, which compromise IFN-γ production after IL-12 stimulation, have been identified to predispose one for Kaposi sarcoma development. Therefore, KSHV co-infection alters the innate immune control of EBV by NK cell differentiation and elicits weaker adaptive immune responses against itself compared to EBV. Furthermore, KSHV-specific immune control relies on different T cell subsets and functions compared with protective immune responses against EBV. 

## 5. Human Papilloma Virus (HPV)

Both EBV and human papillomavirus (HPV) are associated with carcinomas of the oropharyngeal and nasopharyngeal cavities [5]. Non-keratinizing nasopharyngeal carcinoma (NPC) and oropharyngeal squamous cell carcinoma (OPSCC) are nearly uniformly associated with EBV or associated with HPV by up to one-third, respectively [85,86]. Mainly HPV16, but at a lower percentage also HPV18, HPV33, HPV35 and HPV58, can be found in OPSCC [87,88]. In 11% of OPSCCs, both EBV and HPV can be detected [89]. However, it remains unclear what percentage of the respective tumor cells harbors both viruses and if co-infection contributes to tumorigenesis in the oropharyngeal and nasopharyngeal cavities. 

HPV infection of the stratified epithelium precedes OPSCC [90] and it has been suggested that epithelial cells require preconditioning, for example, by mutations in the NF-κB pathway, of HLA genes, and in chromatin remodeling, in the PI3K/MAPK pathway of RAS and of p53, to allow EBV to establish a latency IIa infection and drive NPC tumorigenesis [85]. It is tempting to speculate that HPV could substitute for some of these preconditioning lesions. Along these lines, ephrin receptor A2 (EphA2) has been argued to be the main entry receptor for EBV into epithelial cells [91,92], and the E6 gene product of HPV18 was found to up-regulate EphA2 on immortalized esophageal carcinoma cells [93] (Figure 1D). Benign cervical cells even up-regulated EphA2 15-fold after HPV18 E2 expression [94], and EphA2 was also up-regulated in HPV-associated cervical carcinoma [95]. Thus, HPV infection in the oropharyngeal and nasopharyngeal cavities could facilitate the EBV infection of stratified epithelial cells. 

In addition, HPV infection could also facilitate latent gene expression after EBV entry to promote latency-IIa-driven NPC development (Table 1). In organotypic rafts of tonsillar epithelium, HPV infection seemed to reduce EBV replication (Figure 1D) and down-regulated the expression of early lytic EBV genes, including BZLF1, BRLF1, BMRF1 and BALF5, but up-regulated expressions of the small, non-translated viral RNA EBER1 and the unchanged expression of the latent EBV gene product EBNA2 [96]. HPV E7 seemed to be sufficient for this suppression of lytic EBV replication [96] and is known to down-regulate the retinoblastoma (Rb) gene product [97]. However, Rb is required for lytic EBV replication in both differentiated epithelial and B cells, and HPV E7 may therefore compromise lytic EBV replication by down-regulating Rb [98]. Furthermore, transgenic HPV16 E6 and E7 expression and EBV infection of hypopharyngeal cancer cells and of normal oral keratinocytes led to LMP2 expression and caused elevated invasion capacity [99]. Thus, HPV co-infection may suppress lytic replication and lead to enhanced tumorigenesis due to latent EBV infection in epithelial cells. 

## 6. Human Cytomegalovirus (HCMV) and Other Infections

Persistent EBV infection also interacts with other co-infections, but presumably, these are not harbored in EBV-infected cells. These include human cytomegalovirus (HCMV) and *Helicobacter pylori* infections. The β-herpesvirus HCMV establishes persistent infections in the myeloid hematopoietic lineage and endothelial cells, and its periodic reactivation from this latency pool is associated with immune activation and indicative of loss of immune control [100]. Curiously, HCMV and EBV, despite being both herpesviruses, drive the development of lymphocyte compartments into opposite directions (Figure 1E). This is exemplified by NK and CD8^+^ T cells. HCMV infection is known to drive adaptive NKG2C^+^ NK cell accumulation [101,102]. These are terminally differentiated and express killer-immunoglobulin-like receptors (KIRs) and Leukocyte-immunoglobulin-like receptor subfamily B member 1 (LILRB1/LIR1/ILT2/CD85j). Furthermore, they express the low-affinity Fc receptor CD16 and are probably particularly efficient in antibody-mediated cellular cytotoxicity (ADCC) [103,104]. In contrast, early differentiated NKG2A^+^ NK cells expand during the symptomatic primary EBV infection, called infectious mononucleosis (IM) [54,105]. These preferentially control lytic-EBV-replicating B cells [22,54,106,107]. NK cell differentiation decreases their potential to control EBV infection [22,52,53,108,109]. Similar to early NK cell expansion due to EBV and terminally differentiated NK cell accumulation due to HCMV infection, CD8^+^ T cells that recognize these two viruses accumulate at opposite ends of their differentiation [110,111]. HCMV-specific CD8^+^ T cells are often CD45RA re-expressing TEMRA cells that have lost all early co-stimulatory molecules like CD27 and CD28, but frequently express the senescence markers CD57 and KLRG1 [111]. In contrast, EBV-specific CD8^+^ T cells express CD27 and CD28, as well as often CD127 and CD45RO [111]. The terminal differentiation of HCMV-specific NK and CD8^+^ T cells may be driven by virus-infected myeloid cells, while EBV-infected B cells may promote early differentiated NK and CD8^+^ T cell accumulation. However, are these alternative lymphocyte differentiation patterns just co-existing or do they determine susceptibility to diseases? HCMV-driven NK cell differentiation could abolish immune protection against PTLD [108]. Accordingly, several studies have observed an increased risk to develop PTLD upon the reactivation of HCMV (Table 1) [112,113,114]. Furthermore, HCMV co-infection can increase the severity of IM and hemophagocytic lymphohistiocytosis [115,116,117]. Finally, HCMV and EBV reactivation leads also to increased graft damage and rejection in renal transplant patients [118,119]. In addition to their interaction during EBV-associated malignancies and after transplantation, HCMV and EBV could also interact during autoimmune diseases. Along these lines, it was found that EBV infection, IM and elevated EBV-specific antibody responses increased the risk for multiple sclerosis (MS) [120,121,122], while HCMV infection had a slightly protective effect [120,122]. However, these opposing effects in MS are so far not mechanistically understood. Nevertheless, EBV and HCMV infections shape human lymphocyte compartments in opposing directions and this may have opposing effects on human pathologies.

The gastric epithelium has been suggested to be shaped by *Helicobacter pylori* (*H. pylori*) and EBV co-infection during gastric carcinogenesis [123,124,125,126,127]. A total of 10% of gastric carcinomas are EBV-associated and constitute around one-third of the 300,000 new EBV-associated malignancies every year [6]. *H. pylori* and in particular cytotoxin-associated gene-A (CagA)-expressing strains have been long associated with gastric cancer [5,128]. Three mechanisms have been discussed for the collaboration of *H. pylori* and EBV in promoting gastric carcinoma formation. These are additive persistent inflammation, synergistic epigenetic modifications and reciprocal support of bacterial and viral pathogenicity factors during co-infections. *H. pylori* and EBV co-infections have been found to be associated with more severe gastritis in children [129]. Especially IL-1β and associated IL-8 and TNFα production were found to be elevated due to the co-infection of EBV and *H. pylori* [130]. The respective inflammation but also EBV infection of gastric epithelial cells are associated with aberrant DNA methylation, silencing tumor suppressors, including E-cadherin, p73 and CDKN2A [131]. Finally, *H. pylori* induces, via its CagA protein, an effector molecule of its type IV secretion system, IL-8 to attract EBV-infected B cells [132]. Epithelial cell infection via contact with EBV-producing B cells is indeed 1000-fold more efficient than that via the free virus [133,134]. *H. pylori* induces lytic EBV replication in gastric epithelial cells [135], and EBV viral loads are elevated in *H. pylori*-infected individuals with gastroduodenal diseases [136]. Therefore, elevated lytic EBV replication in the *H. pylori*-colonized gastric epithelium could increase EBV infection of gastric epithelial cells (Figure 1F and Table 1). In these, EBV infection causes SHP1 promotor hypermethylation, preventing *H. pylori* CagA inactivation and facilitating CagA-mediated SHP2 deregulation during gastric carcinogenesis [137]. Therefore, Helicobacter pylori could facilitate gastric epithelium infection with EBV, cooperate in elevated inflammation and benefit from EBV-induced epigenetic modifications for both cellular transformation and the more efficient effector function of its type IV secretion system in virus-infected epithelial cells. However, in order to assess if these interactions contribute to gastric carcinogenesis, a dual infection model that could allow for the manipulation of these factors and support gastric carcinoma formation would be required. 

## 7. Conclusions and Outlook

The high prevalence of persistent EBV infection in the adult human population implies that all other infections occur as co-infections. Some of these seem to increase EBV-associated pathogenesis. This is well-established for HIV, *Plasmodium falciparum* and KSHV, and more speculative for HPV, HCMV and *Helicobacter pylori*. Co-infection models that could determine the respective EBV-associated pathologies are just starting to be established and could be informative for unravelling the mechanisms of EBV-associated lymphomagenesis and carcinogenesis, as well as for the identification of crucial components of EBV-specific immune control. This information could then be used to develop interventions that could treat or prevent the pathogenic collaboration of the respective co-infections. 

Along these lines, interfering with HIV infection using ART has nearly abolished some AIDS-associated EBV-positive B cell lymphomas, such as DLBCL and primary CNS lymphoma, but has failed to significantly lower the incidence of Burkitt’s and Hodgkin’s lymphomas in people living with HIV (PLWH) [15]. Either an even more efficient suppression of HIV replication is required to achieve protection from the associated Burkitt’s and Hodgkin’s lymphomas or therapeutic vaccination to elicit T cell responses that target the EBV antigens that are expressed in these tumors (EBNA1, LMP1 and LMP2) could be considered [33]. For Sub-Saharan endemic Burkitt’s lymphoma, it will be interesting to characterize if the R21/Matrix-M vaccine against malaria, only the second such vaccine recommended by the WHO, will have an impact on Burkitt’s lymphoma incidence [138]. Otherwise, also in this setting, therapeutic vaccination against EBNA1 may efficiently target these mostly childhood tumors. KSHV has been proposed to be a promising vaccine target on its own, possibly reducing high-incidence geographical regions such as Sub-Saharan Africa into low-incidence areas [139]. However, it remains less clear which antigens one would choose to target or prevent KSHV infection in B cells. For PEL, an EBNA1-based vaccine would also be beneficial to target the 90% of this tumor entity that also harbors EBV, mostly in latency I. An HCMV-specific vaccine has also been pursued for more than 50 years, but so far without success [140]. However, what we can learn from these efforts is that, most likely, different antigens and different vaccine formulations need to be combined to elicit antibody, CD4^+^ and CD8^+^ T cell responses. While T cell responses are probably even more important in EBV-specific immune control, also for EBV-specific vaccination, a combination of CD4^+^ and CD8^+^ T cell antigens and vaccination strategies that stimulate both T cell populations may be crucial [141], with an additional challenge to keep the elicited T cell responses in an early differentiation stage that is characteristic of EBV-specific T cells in healthy virus carriers [110,111]. Finally, treatments against HPV and *H. pylori* already exist in the form of efficient vaccines and strong acid suppression together with antibiotics and/or bismuth, respectively [142,143]. However, the respective EBV-associated nasopharyngeal and gastric carcinomas, the development of which may be supported by HPV and *H. pylori*, respectively, may also be targeted by vaccinations to elicit protective T cell responses against latency I/IIa antigens, primarily EBNA1 and LMP2. Thus, for some EBV-associated pathologies, a better understanding of the contribution of co-infections may allow for the development of therapeutic strategies that target both EBV and the co-infecting pathogen.

Even so, the mechanisms by which co-infections lead to EBV-associated pathologies remain poorly defined but may be further characterized in new experimental models and innovative approaches, such as the co-infections of B cells with KSHV and EBV, and humanized mice. The existing (HPV) and emerging vaccines (EBV, KSHV and malaria) will provide the best insights into the importance of co-infections for EBV-associated pathologies. The gain of knowledge and efficacy of targeting both EBV and co-infections that increase pathologies may only be limited by how widely these new vaccines will be distributed, especially in the developing countries of Sub-Saharan Africa, in which many of the co-infection-associated EBV pathologies occur.

## Figures and Tables

**Figure 1 cancers-15-05739-f001:**
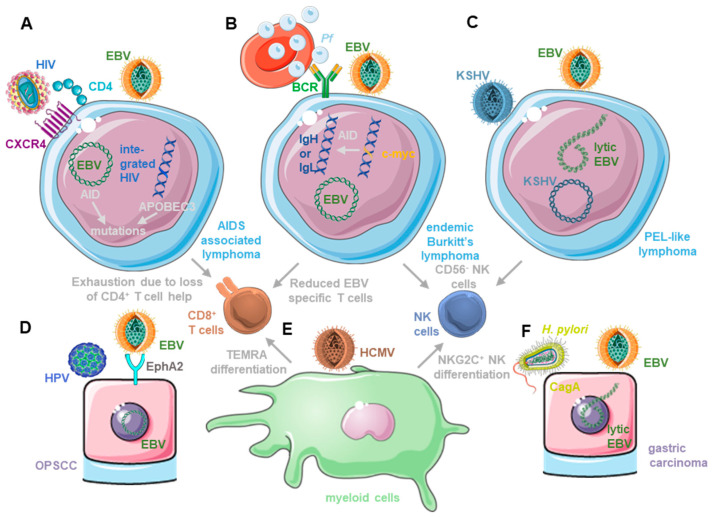
Co-infections influence both EBV-associated tumorigenesis and its immune control. (**A**) EBV infection renders B cells susceptible to HIV infection and both contribute to mutagenesis by up-regulating-activation-induced cytidine deaminase (AID) and APOBEC3. In addition, HIV compromises EBV-specific immune control by CD8^+^ T cell exhaustion after depletion of CD4^+^ T cell help. (**B**) *Plasmodium-falciparum* (*Pf*)-infected erythrocytes do not only stimulate EBV-infected B cell activation, which might facilitate AID-dependent c-myc translocation, but the ensuing endemic Burkitt’s lymphoma is also associated with diminished EBV-specific T cell immunity and CD56^−^ NK cell differentiation with limited ability to control lytic EBV replication. (**C**) KSHV and EBV co-infection causes primary effusion lymphoma (PEL), in which KSHV drives plasma cell differentiation, thereby increasing lytic EBV replication. Double-infection also causes terminal CD56^−^ NK cell differentiation that compromises the innate immune control of lytic EBV replication. (**D**) HPV and EBV co-infection can be found in around 10% of oropharyngeal squamous cell carcinomas (OPSCC). HPV up-regulates the entry receptor for EBV into epithelial cells and the ephrin A2 receptor (EphA2) and dampens lytic EBV infection, thereby possibly supporting transformation of a latent EBV infection in epithelial cells. (**E**) HCMV infection causes terminal CD8^+^ T cell (CD45RA re-expressing TEMRA cells) and NK cell differentiation (adaptive NKG2C^+^ NK cells). EBV requires early differentiated CD8^+^ T cells and NK cells for efficient immune control. (**F**) Cytotoxin-associated gene A (CagA) expressing *Helicobacter pylori* (*H. pylori*) is associated with the majority of gastric carcinomas. An amount of 10% of this tumor is latently EBV-infected. *H. pylori* induces lytic EBV replication, and EBV impairs the inhibitory dephosphorylation of CagA, enhancing its tumorigenic potential.

**Table 1 cancers-15-05739-t001:** Co-infections that are associated with EBV-driven pathologies.

Co-Infecting Agent	Associated EBV Pathology	Possible Mechanism
HIV	EBV-associated lymphomas	Compromised EBV-specific immune control
*Plasmodium falciparum*	Burkitt’s lymphoma	Stimulation of EBV-infected B cells to acquire c-myc translocation
KSHV	Primary effusion lymphoma	Co-infection in the transformed plasmablasts
HPV	Nasopharyngeal carcinoma	Support of transforming latent EBV infection
HCMV	Post-transplant lymphoproliferative disease	Decreased EBV-specific immune control
*Helicobacter pylori*	Gastric carcinoma	Facilitation of EBV infection of gastric epithelial cells

## Data Availability

Data are contained within the article.

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
