# Peer review of "Modulation of Epstein-Barr-Virus (EBV)-Associated Cancers by Co-Infections"

_cancers, 2023, doi:10.3390/cancers15245739_

Round 1

Reviewer 1 Report

Comments and Suggestions for Authors

At present manuscript submitted by MÏ‹nz is highlighted on the modulation of Epstein Barr virus (EBV) associated cancers by co-infections. It is presented novel theory for the EBV infection and clinical prevented strategy.

The authors should be presented the innovation and limitation.

Comments on the Quality of English Language

The language is right.

Author Response

I thank this reviewer for his/her suggestions. In response to these I have now incorporated a new paragraph in the conclusions and outlook section of the revised manuscript. In this paragraph I highlight that the emerging new vaccines against EBV, KSHV and malaria will provide the best insights into the importance of co-infections for EBV associated pathologies. However, as a limitation the efficacy with which these new prophylactic treatments will be rolled out in developing countries where most of the co-infection associated pathologies occur, will determine how much we can learn from them. This new paragraph is inserted on page 9 of the revised manuscript.

Reviewer 2 Report

Comments and Suggestions for Authors The author describes several situations of co-infections of viral and bacterial pathogens and the molecular mechanisms linked to them. The article would improve in quality if it could better explain (i) EBV CMV interactions, in particular possible interactions in immunocompromised subjects (ii) in general, the consequences in human clinical practice. For instance, The author should put more emphasis on diseases/disorders linked to EBV CMV coinfections in the context of immunodepression. cf Sánchez-Ponce Y, Fuentes-Pananá EM. The Role of Coinfections in the EBV-Host Broken Equilibrium. Viruses. 2021 Jul 19;13(7):1399. doi: 10.3390/v13071399. PMID: 34372605; PMCID: PMC8310153. i.e. EBV and CMV together were associated with more severe forms of infectious mononucleosis, hepatitis, and hemophagocytic lymphohistiocytosis. Indeed, CMV reactivation has been demonstrated to induce immunosuppression, which predicts EBV reactivation and PTLD risk. EBV reactivation concurrent with CMV is associated with gammapathy in renal transplanted patients (Drouet E, Chapuis-Cellier C, Bosshard S, Verniol C, Niveleau A, Touraine JL, Garnier JL. Oligo-monoclonal immunoglobulins frequently develop during concurrent cytomegalovirus (CMV) and Epstein-Barr virus (EBV) infections in patients after renal transplantation. Clin Exp Immunol. 1999 Dec;118(3):465-72. doi: 10.1046/j.1365-2249.1999.01084.x. PMID: 10594569; PMCID: PMC1905451......

Author Response

I thank this reviewer for pointing out these clinical observations. I have now extended the discussion of EBV and HCMV co-infection to incorporate the two references that were cited by this reviewer, as well as additional studies that document a higher incidence of PTLD upon HCMV co-infection, more severe forms of IM and HLH in the presence of both viral infections, and more frequent renal transplant damage upon re-activation of both EBV and HCMV. These findings are discussed on page 7 of the revised manuscript.

Reviewer 3 Report

Comments and Suggestions for Authors

Excellent and succint review. Well written with good summary sentences. References are adequate and well sourced. No major comments.

Author Response

I thank this reviewer for his/her encouraging comments.